# Endothelial Dysfunction with Aging: Does Sex Matter?

**DOI:** 10.3390/ijms252212203

**Published:** 2024-11-13

**Authors:** Jakub Jozue Wojtacha, Barbara Morawin, Edyta Wawrzyniak-Gramacka, Anna Tylutka, Ana Karyn Ehrenfried de Freitas, Agnieszka Zembron-Lacny

**Affiliations:** 1Department of Applied and Clinical Physiology, University of Zielona Gora, 28 Zyty Str., 65-417 Zielona Gora, Poland; jakubwojtacha1@gmail.com (J.J.W.); b.morawin@cm.uz.zgora.pl (B.M.); e.gramacka@cm.uz.zgora.pl (E.W.-G.); a.tylutka@cm.uz.zgora.pl (A.T.); 2School of Health Science, Positivo University, 5300 Professor Pedro Viriato Parigot de Souza Street, Campo Comprido, Curitiba 81280-330, PR, Brazil; akarynfreitas@gmail.com; 3Department of Cardiology, Hospital da Cruz Vermelha Brasileira Filial do Paraná, Av. Vicente Machado, 1280, R. Cap. Souza Franco, 50-Batel, Curitiba 80420-011, PR, Brazil

**Keywords:** blood cell count, endothelial progenitor cells, nitric oxide, 3-nitrotyrosine, systemic inflammatory index, older adults

## Abstract

Oxidative stress and inflammation accompany endothelial dysfunction that results from the excessive or uncontrolled production of reactive oxygen and nitrogen species (RONS) in older adults. This study was designed to assess the usefulness of serum oxi-inflammatory component combinations in vascular disease prediction and prevention with regard to sex. Women (*n* = 145) and men (*n* = 50) aged 72.2 ± 7.8 years participated in this project. The females demonstrated the elevated production of hydrogen peroxide (H_2_O_2)_ and nitric oxide (NO) responsible for intravascular low-density lipoprotein oxidation. NO generation was enhanced in the women, but its bioavailability was reduced, which was expressed by a high 3-nitrotyrosine (3-NitroT) concentration. The relation of NO/3-NitroT (r_s_ = 0.811, *p* < 0.001) in the women and NO/3-NitroT (r_s_ = −0.611, *p* < 0.001) in the men showed that sex determines endothelial dysfunction. RONS generation in the women simultaneously promoted endothelial regeneration, as demonstrated by a ~1.5-fold increase in circulating progenitor cells. Inflammation-specific variables, such as the neutrophil-to-lymphocyte ratio, the systemic immune inflammation index, and the neutrophil-to-high-density lipoprotein (HDL) ratio, were reduced in the women and showed their diagnostic utility for clinical prognosis in vascular dysfunction, especially the C-reactive-protein-to-HDL ratio (AUC = 0.980, specificity 94.7%, sensitivity 93.3%, OR = 252, 95% CI 65–967, *p* < 0.001). This study is the first to have revealed sex-specific changes in the oxi-inflammatory response, which can generate the risk of cardiovascular events at an older age.

## 1. Introduction

Endothelial dysfunction is a crucial factor preceding the development of cardiovascular diseases (CVDs). It is generally complicated by simultaneous oxidative stress and inflammation in the senescence of vascular cells [1]. The risk factors of augmented cardiovascular aging, such as a low physical activity level and a high visceral fat content, can induce changes in endothelium morphology and metabolism, which contributes to arterial stiffness, hypertension, atherosclerosis, stroke, and coronary artery disease in older adults [2,3,4]. In this regard, aging is a major risk factor and represents a crucial health challenge in the pathogenesis of CVDs [5]. According to the American Heart Association Report on Heart Disease and Stroke Statistics in 2024, the incidence of CVD was typically 35–40% in people aged between 40 and 60 years, 77–80% in patients between 60 and 80 years of age, and over 85% in patients over the age of 80. CVDs have been considered the true diseases of aging [6,7]. In Europe, more than 4 million people die from cardiovascular CVDs each year [8]. The incidence of CVDs in Poland is also on the rise. At present, CVDs account for 38% of all deaths, with ischemic heart disease being the principal cause [9]. The Polish population is aging, as indicated by a life expectancy of 73 years in men and 81 years in women [10]. The disparity between sexes is among the highest in Europe and may be related to men’s six-fold higher incidence of accidental death, 10% less use of medical services, and greater exposure to modifiable risks [9].

In several studies on the impact of aging, the parameters obtained from direct measurements of the heart and vascular structures or functions have been used as either specific factors for association analysis, or collective predictors of future CVD risk [4]. Aging research includes analyses of numerous cardiac and vascular functional or structural parameters, such as coronary artery calcification, the arterial intima-media thickness of carotid arteries, pulse wave velocity, and flow-mediated vasodilation. The rationale behind their selection is that these parameters can serve as more quantifiable phenotypes to measure the aging process for the cardiovascular system, and it is also governed by underlying aging [7]. However, there is a lack of integrated markers and scores to measure cardiovascular age based on an aging-specific profile and functional measurements that can be applied in clinical practice. Cardiovascular markers are often deployed without assessing gender-specific cut-off values. It is now becoming apparent that the use of such cut-off values can improve prognostication and discrimination in some clinical situations [11].

Regular blood tests to evaluate the inflammatory process status are useful in the early diagnosis of several diseases, as well as in the clinical prognosis of CVDs. In particular, the complete blood count profile is readily available, inexpensive, and provides information regarding various cell types, i.e., leucocytes, neutrophils, monocytes, lymphocytes, and platelets. Recently, some studies have found that the combination of oxi-inflammatory factors and lipoproteins, such as the neutrophil-to-lymphocyte ratio (NLR), the lymphocyte-to-monocyte ratio (LMR), the platelet-to-lymphocyte ratio (PLR), the systemic immune inflammation index (SII), and the C-reactive-protein-to-high-density-lipoprotein ratio (CHR) are applicable prognostic markers in patients with a variety of cancers [12,13,14], infectious diseases [15,16], obesity [17,18], and diabetes [19], as well as coronary artery disease [20,21,22,23] and acute ischemic stroke [24,25]. These easily measured parameters are routinely calculated in daily practice as a part of the complete blood count report and basic biochemical analysis [26].

Inflammation has been implicated in age-associated endothelial dysfunction and the worsening of large artery stiffness. Age-related inflammation, called inflammaging, is characterized by increased levels of circulating C-reactive protein (CRP), as well as pro-inflammatory cytokines, including interleukin (IL)-1, IL-2, IL-6, IL-8, IL-12, IL-13, IL-15, IL-18, IL-22, IL-23, tumor necrosis factor (TNFα), and interferon (IFNγ) [27]. These circulating markers of inflammation, CRP in particular, have been positively related to aortic stiffness and inversely related to endothelium-dependent dilation in older adults [28]. Although some inflammatory mediators are derived from vascular cells per se, immune cells infiltrating the arterial adventitia have also been indicated as an important source of both inflammatory cytokines and reactive and nitrogen species (RONS), such as hydrogen peroxide (H_2_O_2_) and nitric oxide (NO). This is supported by data that show the increased accumulation of neutrophiles and lymphocytes T in the adventitia of aged arteries and the role of adventitial-derived RONS in arterial inflammation in the setting of CVDs [7,29,30]. Our earlier study demonstrated that aging is associated with profound changes in the immune risk profile expressed as the lymphocyte CD4-to-CD8 ratio, also pointing to an interplay between the immune cells and the endothelial cells in older age [30]. However, it did not demonstrate sex-related changes for inflammatory or endothelium-specific variables. On the basis of the assembled data on oxidative stress and inflammation and their complications, including the changes in endothelium metabolism from subclinical dysfunction to revealed disease, this study was designed to assess the prognostic values of serum oxi-inflammatory factors in endothelium metabolism depending on sex.

## 2. Results

### 2.1. Study Population

The body mass index (BMI) ranged from 18.6 to 38.8 kg/m^2^ in the women and from 19.9 to 40.8 kg/m^2^ in the men. Approximately 27% of the studied seniors had a normal body mass (18.5–24.9 kg/m^2^), and 73% were classified as overweight or obese (≥25 kg/m^2^). A high fat mass dominated among the women (women 24.2 ± 6.1 kg, men 20.4 ± 6.7 kg), whereas a high fat-free mass was characteristic of the men (women 43.9 ± 4.5 kg, men 59.0 ± 7.2 kg). However, 80% of the men showed an excessive content of visceral fat (>12 VF unit), whereas only 7% of the women had 12 < VFC ≤ 14. Significant differences in visceral fat content were observed between the individuals with low-grade inflammation CRP < 3 mg/L (VFC 10.7 ± 3.4) and high-grade inflammation CRP ≥ 3 mg/L (VFC 12.5 ± 3.7). Systolic blood pressure > 140 mmHg was recorded in 67%, and diastolic blood pressure < 70 mmHg was found in 13% of the studied seniors, which increases the cardiovascular risk in older patients, according to the latest guidelines of the European Society of Hypertension and the European Society of Cardiology [31]. On average, the participants covered the distance of 452 ± 73 m in a 6 min walk test (6MWT) at a gait speed of 1.3 ± 0.2 m/s. Approximately 62% of the subjects achieved a gait speed within the range of 1.0 to 1.3 m/s, which classified them as active, and only 6% achieved a gait speed < 1.0 m/s, which classified them as inactive. The remaining individuals (32%) represented the extremely fit demographic according to classification by Middelton et al. [32]. The results of 6MWT showed that the University-of-Third-Age students represented healthy aging. According to Zielińska-Więczkowska et al. [33], students’ participation in this university clearly determines their knowledge concerning illnesses, afflictions, depression, and the health benefits of physical activity, regardless of sex. There were no significant differences in 6MWT between the women and men (Table 1).

### 2.2. Hematological Variables

The white blood cell count was within the referential range in all our study patients. There were fewer neutrophils, whereas the lymphocytes, monocytes, and platelets tended toward higher values in the women compared to those of the men (Table 2), which resulted in changes in inflammatory indices: the NRL and the SII (Figure 1A,B). The red blood cell parameters were within the referential range according to the Cardiovascular Health Study [34]. However, concentrations of Hb < 13 g/dL in 20% of the women and Hb < 14 g/dL in 60% of the men indicated the existence of anemia in old age [35]. According to Culleton et al. [35], Hb concentrations from 13.0 to 15.0 g/dL for the women and from 14.0 to 17.0 g/dL for the men are the optimal values to avoid hospitalization and mortality in old age.

### 2.3. Lipoprotein–Lipid Profile

The total cholesterol (TC) and lipoproteins have been proven to be the strongest biomarkers of vascular aging [36]. In this study, high levels of triglycerides (TG) > 150 mg/dL were found in 36%, whereas high levels of TC > 200 mg/dL and non-HDL > 130 mg/dL were found in 72% of our study sample. The lipoprotein–lipid profile differed by sex (Table 3). The TG and TC levels were higher by ~20%, and the LDL level was lower in the women compared with that of the men. The particular difference was observed in oxLDL which was two-fold higher in the women than that in the men. In the women, oxLDL was highly correlated with endothelium-specific variables such as H_2_O_2_ (r_s_ = 0.71, *p* < 0.001) and NO (r_s_ = 0.49, *p* < 0.001), but in the men, oxLDL was correlated with 3-nitrotyrosine (3-NitroT) (r_s_ = 0.55, *p* < 0.001). This confirms the earlier findings that oxLDLs could affect endothelial cell activities in almost every aspect, such as proliferation, differentiation, apoptosis, and senescence [30,37]. The TG/HDL ratio in both the sexes reached a value close to two. There are no widely accepted reference ranges for the TG/HDLs. Generally, a value < 2 is regarded as a good result, whereas a ratio of three or more is linked to higher-risk CVDs [23]. However, a significant difference in the TG/HDL ratio was recorded between the individuals with low-grade inflammation CRP < 3 mg/L (TG/HDL 1.82 ± 0.99) and high-grade inflammation CRP ≥ 3 mg/L (TG/HDL 2.24 ± 1.30), thereby indicating an association of the TG/HDL ratio with inflammaging. Thera were significant differences for the TG, the TC, the HDLs, and the non-HDLs, but not for the other lipoprotein–lipid and inflammatory variables in the women and the men stratified based on the CRP levels.

### 2.4. Endothelium-Specific Variables

Elevated levels of H_2_O_2_, a promoter of inflammation in the endothelial and smooth muscle cells [1], were observed in our female sample. Although NO generation was increased in the women, its bioavailability was reduced and demonstrated by increased 3-NitroT concentrations. The relation of NO/3-NitroT was recorded at r_s_ = 0.811 (*p* < 0.001) in the women and at r_s_ = −0.611 (*p* < 0.001) in the men, which confirms the previous recommendations by Bencsik et al. [38] that 3-NitroT could be assessed as a risk factor of endothelial dysfunction, but this relationship varies depending on sex. However, it should be emphasized that significantly more NO was generated (282 ± 52 μmol/L) among the women with CRP < 3 mg/L than that in the women with CRP ≥ 3 mg/L (216 ± 197 μmol/L). This obviously indicates that inflammatory status does not only inhibit NO generation, but also its bioavailability. Substantial changes were observed in the level of circulating endothelial progenitor cells (EPCs) and high-mobility group box 1 (HMGB1), which were ~1.5-fold higher in the women than those of the men (Table 4). EPCs are involved in maintaining appropriate endothelial function, whereas HMBG1 participates in the inflammatory response and attracts progenitor cells to hone into areas of inflammation, promoting the regeneration process [39,40]. Therefore, the analysis of EPCs and HMGB1 could be used in the assessment of the course of endothelial regeneration in the female population and could be more effective if EPCs and HMGB1 are tested jointly at the same time.

### 2.5. Inflammatory Indices

Most observational studies and clinical trials have used high-sensitivity CRP as a biochemical marker of inflammation due to its stability and ease of measurement. In our study, the CRP concentration was within the referential range in 138 individuals (CRP < 3 mg/L) and above the range in 57 individuals (CRP ≥ 3 mg/L), according to the reference values described by Pearson et al. [41]. The CRP values tended to reach lower levels in the women (2.25 ± 1.88 mg/L) than those in the men (2.92 ± 2.32 mg/L), although the differences were not as pronounced as in the other inflammatory indices, such as the NLR, the SII, the NHR, and the CHR (Figure 2). Systemic inflammatory status, expressed as a CRP value, was considerably associated with changes in the NLR, the SII, and the NHR. The individuals with low-grade inflammation CRP < 3 mg/L demonstrated lower values of NLR (2.44 ± 2.01), SII (523 ± 410), and NHR (0.55 ± 0.026) than those of the patients with high-grade inflammation CRP ≥ 3 mg/L (NLR 3.23 ± 2.44, SII 660 ± 445 and NHR 0.074 ± 0.048). Significantly lower levels of NLR (2.29 ± 2.03) and SII (602 ± 537) were observed in the women compared to those of the men (NLR 2.93 ± 2.24 and SII 776 ± 454). The NHR and the CHR were even two-fold lower in the women (0.055 ± 0.030 and 0.031 ± 0.028) than those of the men (0.090 ± 0.047 and 0.075 ± 0.057). The results of ROC analysis for the NLR, the SII, and the NHR as an area under the ROC curve (AUC) reached the value >0.6, indicating an average diagnostic potential for clinical prognosis in older patients with chronic inflammation, in contrast to the CHR, for which a very high value of AUC >0.9 was demonstrated (Table 5). The optimal threshold values corresponded to 1.67 for the NLR (AUC = 0.655, specificity 77.2%, sensitivity 53.3%, OR = 3.87, 95% CI 1.911–7.829), 350 for the SII (AUC = 0.627, specificity 86.0%, sensitivity 34.1%, OR = 3.17, 95% CI 1.386–7.244), and 0.064 for the NHR (AUC = 0.636, specificity 52.6%, sensitivity 77.6%, OR = 3.82, 95% CI 1.991–7.451). The highest changes and values of AUC = 0.980 and OR = 252 were observed for the CHR (Figure 2 and Table 5), which is indicative of a very high probability of endothelial dysfunction once the cut-off value (0.038) has been exceeded. Moreover, the highest specificity (94.7%) and sensitivity (93.3%) observed for the CHR indicates a low level of false positive results during diagnostic procedure using the CHR. Therefore, it seems reasonable to assess the combinations of CRPs and HDLs together to discriminate between healthy subjects and patients with endothelial dysfunction and CVD risk, including atherosclerosis.

## 3. Discussion

Aging has been regarded as an unmodifiable CVD risk factor at both the individual and population levels. Therefore, although age is an essential factor considered in all the available CVD risk-assessment tools used in clinical practice, the recommendations specifically made on how to lower the aging-related CVD risk have been very limited as a part of the prevention guidelines for CVDs. Most research concerning the CVD burden at the population level used only age-standardized rates, and the CVD burden caused by the sex-specific impact of aging has been largely ignored [4,11]. This leads to an important question whether it is possible to diminish the risk of age-related CVDs without taking into account sex. The sex differences in body composition have been well established [42], and the differences we observed in this study were hardly surprising as they were consistent with our previous observations [43,44]. The women had a higher fat mass (FM) than the men, and the men had a higher fat-free mass (FFM) than the women. As the BMI increased in both the sexes, the body fat percentage remained higher in the women than that of the men (Table 1). Importantly, the differences between the two sexes are not only in the percentage of total body fat, but also in its distribution on different parts of the body [45]. The visceral fat content (VFC) was excessive in 80% of the men, leading to increased inflammation. One of possible underlying mechanisms is that the growth of adipose tissue and the infiltration of immune cells lead to an increase in pro-inflammatory cytokines, which causes an enhanced systemic inflammatory response exhibited by high C-reactive protein levels [44]. A significant difference in VFC was detected between the individuals with low-grade inflammation CRP < 3 mg/L (VFC 10.7 ± 3.4) and high-grade inflammation CRP ≥ 3 mg/L (VFC 12.5 ± 3.7). The available cohort studies have shown that chronic inflammation is involved in an alteration in nutritional status; cognitive decline; a poor physical performance; and neurological, cardiological and vascular events [46]. Our study convincingly showed that sex-specific fat distribution was associated with inflammatory status. In the male sample, all the inflammatory markers, such as the NLR, the SII, the NHR, and the CHR, were found to be elevated. However, the association of visceral fat with metabolic abnormalities is not confined to male populations, as abdominal visceral fat has also been reported as a strong predictor of mortality in obese women [47]. Consequently, it is important to understand the pathogenesis of abdominal visceral fat and its association with metabolic complications [43]. Lipoproteins and lipid particles with triglycerides are the prime source of the fat taken up by adipocytes. As women and men have different intestinal lipoproteins, this could potentially determine which body fat depot the dietary fat will be stored in. Other studies have shown that approximately 27% of ingested fat is stored as intra- and retroperitoneal fat in men [48]. In contrast, approximately 5% of the ingested fat is stored as intraperitoneal fat in women [49]. These studies support the notion that this sex difference in regional body fat distribution is primarily determined by fat uptake rather than lipolysis, especially in old age [50]. The available clinical studies indicate that women have higher levels of HDL and lower levels of TG than men. In a group of 32 normolipidemic older men and postmenopausal women, the mean HDL level was 23% higher, and the TG level was 27% lower in the women compared with that of the men [51]. The absence of testosterone increases the lysosomal degradation of LDL receptors, resulting in an increase in circulating LDL levels [52]. In our study, both the HDL and TG levels were significantly more elevated, whereas the LDLs tended to lower values in the women compared with that of the men. This direction of changes in the lipoprotein–lipid profile could be caused by group heterogeneity (normolipidemic vs. hyperlipidemic individuals) and hypolipidemic drugs (taken by 14% of this study sample). The TG/HDL ratio in both our sex groups reached a value close to two. Although there are no widely accepted reference ranges for the TG/HDL ratio, generally, a value < 2 is considered as a good result, whereas ≥3 is linked to higher-risk CVDs [23]. However, according to the European Society of Cardiology, values of the TG/HDL ratio > 1.65 for women and TG/HDL > 2.75 for men are found to be highly predictive of a first coronary event regardless of the BMI. The high TG/HDL ratio increased the risk of a first coronary event by 50%, meanwhile the LDL level was associated with a more moderately increased risk [53]. The TG/HDL ratio and the LDL values recorded in our study indicated a higher risk of cardiovascular events in the studied women compared to those of the men.

However, the most significant differences were observed in the oxidized low-density lipoprotein (oxLDL) levels, which were two-fold higher in the women, which provides evidence for the sex-specific changes in oxLDL in old age. Previously, high concentrations of oxLDL have been reported for postmenopausal women when compared to those of fertile women and for men with testosterone deficiency [54,55,56,57]. OxLDL promotes foam-cell formation, activates the proinflammatory pathways, and induces smooth-muscle-cell migration, apoptosis, and cell death. One of the major receptors for oxLDL is lectin-like oxLDL receptor (LOX-1), which is upregulated in CVDs, including atherosclerosis. LOX-1 activation in the endothelial cells promotes endothelial dysfunction and induces pro-atherogenic signaling, leading to atherosclerosis. The binding of ox-LDL to LOX-1 encourages the generation of reactive oxygen and nitrogen species, which induce LOX-1 expression and further LDL oxidation [58]. Our studied women were characterized by increased RONS production, which was highly correlated with oxLDL (H_2_O_2_/oxLDL r_s_ = 0.71, *p* < 0.001; NO/oxLDL r_s_ = 0.49, *p* < 0.001). The level of LOX-1 expression in the endothelial cells is very low, but can be promptly induced in the setting of chronic inflammatory conditions [37]. Previously, we demonstrated increased oxLDL in patients with CRP ≥ 3 mg/L, and they were highly correlated with endothelium-specific variables, even though the total cholesterol, LDL, and HDL levels did not differ between the groups [30]. This confirms that the measurement of oxLDL could be used as an early marker for pathological processes in the vascular endothelium [37]. The serum oxLDL level has already been shown to be elevated in patients with established coronary artery disease [59]. In the Edinburgh Artery Study, every 1 μmol/L increase in lipid peroxide was associated with a 17% increase in the risk of atherosclerosis [60]. The study of Bruneck with 1510 individuals aged 40–79 years determined the antibodies against oxLDL in more than 90% of the participants [61]. OxLDL may provide valuable information for the diagnosis and monitoring of CVDs, but the relationship between lipid metabolism and sex requires further study that will also assess the effectiveness of hypolipidemic therapies in relation to sex.

A healthy endothelium is defined by the ability to produce adequate levels of nitric oxide. Reactive oxygen species can affect NO availability and lead to vascular disorders due to compromised NO functionality. An excess of superoxide anion reacts with NO to produce peroxynitrite (ONOO), which leads to comparable redox changes in other oxidants, but is regarded as 1000-fold more potent in causing these redox reactions [62]. ONOO, in turn, can “uncouple” endothelial NO synthase to become a dysfunctional superoxide-generating enzyme that contributes to vascular oxidative stress [63]. Without superoxide, the formation of ONOO by a NO reaction with oxygen is minimal. NO and superoxide do not have to be produced within the same cell to form peroxynitrite because NO can readily move through the cellular membranes and between cells [64]. In the present study, low NO bioavailability was demonstrated by a higher concentration of 3NitroT in the women compared to that of the men. Sex-specific differences in the mechanism of endothelium dysfunction were observed for the relationship between NO and 3-NitroT. In the women, the relation of NO/3-NitroT was recorded at r_s_ = 0.811 (*p* < 0.001), whereas in the men, NO/3-NitroT was inverse r_s_ = −0.611 (*p* < 0.001). Protein nitration can have a remarkable impact on protein structure and function. This has been observed in various pathological conditions associated with oxidative stress and is used as a clinical marker of CVDs [65,66]. Indeed, the tyrosine nitration of proteins, including apolipoprotein A-1, apolipoprotein B-100, and fibrinogen, has been found in plasma collected from patients with coronary artery diseases, suggesting that changes in the function of some nitrotyrosine-modified proteins can create atherosclerotic endothelial dysfunction [67]. NO is the most important nitrosative compound, and its reduced bioavailability is crucial in endothelial dysfunction, and also in promoting arterial remodeling through several mechanisms [68]. Thus, the found changes in NO, H_2_O_2_, and 3-NitroT here can be associated with endothelial dysfunction and can endorse atherogenesis. The repair of a damaged endothelium is dependent on the timely suppression and containment of inflammation; this process is complemented by the activation of endothelial progenitor cells that restore tissue integrity [69]. EPC mobilization from the bone marrow is mainly triggered by inflammation [70]. As such, Morishita et al. [71] investigated the pattern of EPC mobilization and its association with inflammation and oxidative stress markers in patients with CVDs. We observed considerable changes in the circulating EPCs, the level of which was ~1.5-fold higher in the women than that of the men. EPCs are involved in maintaining appropriate endothelial function and have the ability to proliferate, differentiate, and mature into endothelial cells [39]. Women have also demonstrated significantly higher levels of HMGB1, which participates in the inflammatory response and attracts progenitor cells to hone into areas of inflammation [40]. During atherogenesis, HMGB1 is abundantly expressed in endothelial cells, vascular smooth muscle cells, and macrophages [72]. Therefore, HMGB1 expression is upregulated during the process from the lipid deposits to the plaque endothelium. Remarkably, high HMGB1 levels were detected in the serum and tissues of patients with atherosclerosis-related diseases, such as diabetes, ischemic stroke, and hypertension [73]. HMGB1 participates in endothelial dysfunction by binding to the receptor for advanced glycation end products and toll-like receptors, indicating that HMGB1 is highly related to the development of atherosclerosis [74]. However, it has been reported that high concentrations of HMGB1 (≥5 μg/mL), resulting in vascular barrier damage, quickly destroy the junctions between endothelial cells, whereas lower HMGB1 levels (200 ng/mL) cause no destruction of the cell–cell junctions [75,76]. The serum levels of HMGB1 in our study sample were lower (both in the women 50.77 ± 19.15 ng/mL and the men 39.58 ± 9.84 ng/mL) when compared to the values reported by Wolfson et al. [75]. Therefore, the analysis of EPCs and HMGB1 can be used to assess both the extent of damage and the course of endothelial regeneration in inflammaging.

Inflammatory response in older age is a key mechanism in endothelial dysfunction and atherosclerosis progression. The neutrophils secrete inflammatory mediators that can cause vascular wall degeneration. Conversely, the lymphocytes regulate the inflammatory response, and thus play an anti-atherosclerotic role. Therefore, the neutrophil-to-lymphocyte ratio has been proposed as an inflammatory biomarker and a potential predictor of risk and prognosis in CVDs [77,78]. The neutrophil count was significantly lower in our studied women than in the men and resulted in low values of the NLR, the SII, and also the NHR. Moreover, the receiver operating characteristic (ROC) curve analysis of the NLR, the SII, and the NHR resulted in AUC > 0.6, indicating average diagnostic potential for clinical prognosis for older patients with chronic inflammation. According to Belice et al. [79], the NLR was statistically higher in men than women of all ages, including geriatric people, but the ratio did not differ between sexes in younger patients <65 years of age. Zhou et al. [80] demonstrated that increased arterial stiffness in apparently healthy adult males (rather than females) was independently associated with the highest quartile of the NLR. In our previous study, we showed that the progression of immunoaging was dependent on sex [3]. One of the features of immunoaging is a chronic and systemic low-grade inflammation [3], and what promotes inflammation development is a high number of neutrophils. Neutrophils are the main source of interleukin 6, interleukin 8, hepatocyte growth factor, transforming growth factor-β, and matrix metalloproteinases, and they are the factors which play an important role in the development of a variety of older age-related comorbidities [81,82,83]. Neutrophilia inhibits the immune response by suppressing the cytolytic activity of T lymphocytes and natural killer cells, which is expressed in further changes in the NLR. A statistically significant difference was observed between older women and older men in a CD4+-naïve population of T lymphocytes, as well as in CD4CD45RA/CD4CD45RO. A CD4/CD8 ratio < 1 was found in 25% of men, whereas CD4/CD8 ≥ 1 or ≤2.5 dominated in women. CD4/CD8 < 1 is regarded as an immune risk phenotype and can be associated with immunosenescence and chronic inflammatory diseases [84].

Novel diagnostic markers have been sought for several years, with the purpose of increasing the accuracy of predictions for disease progression, patient prognosis, and the probability of cardiovascular incidents. Recently, our investigation, as well as other studies conducted worldwide in this area, have focused on the verification of the predictive potential of combinations of hematologic components integrated into new biomarkers, such as the NLR, the SII, the NHR, and the CHR [85,86]. The results of a large population-based cohort study conducted in China included 6554 Chinese participants and confirmed that a high CHR was a significant risk factor for cardiovascular diseases, new stroke, and heart disease. Seven years of follow-up (2011–2018) showed that 786 people (11.99%) developed cardiovascular diseases. According to the adjusted model, the CHR was also found to be a contributing factor to CVD risk (OR 1.31, 95% CI 1.05–1.64). In addition, a nonlinear relationship was observed between the CHR and the incidence of new cardiovascular diseases, stroke, or cardiovascular problems. Furthermore, stratified analysis detected significant differences between the CHR and age, indicating more adverse effects of a high CHR in younger participants [87]. By and large, medical interventions in patients with increased CHR levels appear to have the potential to reduce the incidence of CVDs and their complications.

Our ROC curve analysis for the NLR, SII, and NHR biomarkers resulted in AUC > 0.6, which indicates medium diagnostic potential for clinical prognosis for older patients with chronic inflammation, as opposed to the CHR, which showed a very high value of AUC > 0.9. Significant differences in the mean and median CHR were observed between the female (0.031 ± 0.028) and male (0.052 ± 0.052) groups (Figure 1D). The highest values of AUC = 0.980 and OR = 289 were observed for the CHR (Figure 1D and Table 5), indicating a very high probability of endothelial dysfunction beyond the cut-off value (0.037). Also, the highest specificity (96.4%) and sensitivity (91.5%) observed for the CHR indicate a very low rate of false positives in diagnosis made using the CHR. Therefore, it seems prudent to assess CRP and HDL associations at the same time to distinguish healthy individuals from patients with endothelial dysfunction and a risk of cardiovascular diseases, including atherosclerosis. A study by Luo et al. [22] suggested that the CHR, as a predictor of coronary artery disease (CAD), has better diagnostic performance than the NLR. The CHR was not only closely associated with the presence and severity of CAD, but was also an independent predictor of severe CAD. This is reflected in the pathophysiology of the CAD process, which mainly involves atherosclerosis. Dyslipidemia and chronic arteritis, as well as various inflammatory factors, play a crucial role in CAD progression. Other studies have shown that the CRP levels and dyslipidemia have a synergistic effect on the pathogenesis of CAD, and the association between dyslipidemia and CAD appears to be enhanced by elevated CRP levels [22]. Cardiovascular diseases are associated with inflammation and abnormal lipid metabolism. However, as confirmed by numerous studies, a single inflammatory index or a singular lipid index cannot accurately predict the prognosis of CVDs, as they are susceptible to the influence of various confounding factors.

## 4. Materials and Methods

### 4.1. Study Population

A total of 220 elderly individuals were recruited between December 2022 and March 2023 at the University of Third Age, which is an organization recommended for elderly people to stay active by participating in many educational programs (Figure 1). The current health status of the participants was assessed on the basis of medical records at a routine follow-up visit to a primary care physician. The inclusion criteria were being ≥60 years of age and attendance at the recruitment meeting. The exclusion criteria included symptoms of acute infection, autoimmune and endocrine diseases, renal failure and tumors, as well as acute coronary disease. Eventually, one hundred and ninety-five individuals (female *n* = 145, male *n* = 50) aged 72.2 ± 7.8 years participated in the project. Both the women and men were distributed into low-grade inflammation (female *n* = 109, male *n* = 29) and high-grade inflammation (female *n* = 36, male *n* = 21) groups based on the measurement of CRP concentration, as a conventional marker of systemic inflammation and cardiovascular diseases according to the reference values (CRP < 3 mg/L or ≥3 mg/L) described by Pearson et al. [41]. Medications taken by the participants included antihypertensive (84%) and hypolipidemic (14%) drugs, as well as anticoagulants, including anti-platelet agents (12%). All the individuals were informed of the aim of this study and signed a written consent to participate in the project. This study protocol was approved by the Regional Bioethics Commission (No. 04/133/2020 and No. UZ/19/2021) in accordance with the Helsinki Declaration.

### 4.2. Body Composition

Body mass and body composition, including VFC, FFM, and FM, were evaluated using a Tanita Body Composition Analyzer MC-980 (Tokyo, Japan) calibrated prior to each test session. Measurements were taken on an empty stomach between 7:00 and 9:00 a.m., and the recurrence of measurement was 98%. Duplicate measurements were made of the studied participants standing upright, and the mean value was included for final analysis.

### 4.3. Blood Sample Collection

Blood samples were collected from the median cubital vein in the morning between 8.00 and 10.00 using S-Monovette tubes (Sarstedt AG & Co. KG, Nümbrecht, Germany). Whole blood samples were placed into tubes containing EDTA and were immediately analyzed. For other biochemical analyses, the blood samples were centrifuged at 3000 rpm for 10 min, and aliquots of serum were stored at −80 °C.

### 4.4. Aerobic Capacity

A 6 min walk test was performed following the standards of the European Respiratory Society and the American Thoracic Society [88]. The object of the test was to walk as fast and as far as possible over a span of six minutes along a marked 30 m walkway with cones placed at regular intervals to indicate the distance covered. The subjects were allowed to self-pace and to rest as needed. The total distance covered was recorded, and the 6MWT gait speed was calculated using the following equation: gait speed (m/s) = total distance(m)/360 s. Following classification by Middelton et al. [32], a gait speed within the range of 1.0 to 1.3 m/s classified the older adults as active, a gait speed < 1.0 m/s classified them as inactive, and a gait speed > 1.3 m/s classified them extremely fit.

### 4.5. Hematological Variables

Hematological parameters, including total white blood cell count (WBC), red blood cell count (RBC), platelet count, differential white cell count (neutrophils, lymphocytes, monocytes, eosinophils, and basophils), and hemoglobin concentration (HB), were determined using a Sysmex XN-1000 (Sysmex Europe Gmbh, Norderstedt, Germany).

### 4.6. Lipoprotein–Lipid Profile

Serum triglycerides (TGs), total cholesterol (TC), high-density lipoproteins (HDLs), and low-density lipoproteins (LDLs) were determined using Biomaxima kits and BM200 Biomaxima (Lublin, Poland). Non-HDL cholesterol was calculated by subtracting the HDLs from the total cholesterol concentration. Oxidized low-density lipoproteins (oxLDL) were determined using ELISA kits from SunRed Biotechnology Company (Shanghai, China), with the detection limit at 30.3 ng/mL. The triglyceride-to-high-density-lipoprotein ratio (TG/HDL) was calculated as TG (mg/L) divided by the HDL level (mg/dL) according to Kosmas et al. [23].

### 4.7. Endothelium-Specific Variables

The NO, 3-NitroT, H_2_O_2_, EPC, and HMGB1 levels were determined using ELISA kits from SunRed Biotechnology Company (Shanghai, China), with detection limits of 2.052 mmol/L, 0.007 nmol/mL, 7.778 ng/mL, 0.125 ng/mL, and 0.526 ng/mL, respectively. The average intra-assay coefficient of variation (intra-assay CV) for the used enzyme immunoassay tests (ELISA) was <5%.

### 4.8. Inflammatory Indices

Serum CRP was measured using a high-sensitivity commercial ELISA kit from DRG International (Springfield Township, Cincinnati, OH, USA), with the detection limit of 0.001 mg/L. The neutrophil-to-lymphocyte ratio (NLR × 10^3^/µL) and the systemic immune inflammation index (SII × 10^3^/µL = (platelets x neutrophils)/lymphocytes)) were calculated and compared to reference values according to Luo et al. [26]. The CHR was calculated as CRP (mg/L) divided by the HDL level (mg/dL), and the NHR was calculated as the neutrophil count (10^3^/µL) divided by the HDL level (mg/dL) [22].

### 4.9. Statistical Analysis

Statistical analyses were performed using R 4.2.1 software [R Core Team. R: A language and environment for statistical computing. R Foundation for Statistical Computing, Vienna, Austria. URL (2022); https://www.R-project.org/, accessed on 18 September 2024]. The variables are described as mean values ± standard deviation (SD) and median (Me).

The expectations for the use of parametric or nonparametric tests were checked using the Shapiro–Wilk and Levene’s tests to assess the normality of distributions and the homogeneity of variances, respectively. Significant differences in the mean values between the women and the men were estimated by one-way ANOVA. If the normality and homogeneity assumptions were violated, the Mann–Whitney nonparametric test was used. Spearman’s rank correlation (r_s_ Spearman’s rank correlation coefficient) was used to explore the relationships between the inflammatory and endothelial variables. The predictive value of inflammatory and endothelial variables was assessed using the receiver operating characteristic curve (ROC). The area under the ROC curve (AUC) was analyzed to provide all the possible classification thresholds. Both univariate and multivariate logistic regression models were used. The optimal threshold value (cut-off) for clinical stratification was obtained by calculating the Youden index. The odds ratio (OR) was determined for univariate analyses. Statistical significance was set at *p* < 0.05.

## 5. Conclusions

There is evidence to support the existence of sex-dependent differences in oxi-inflammatory response. Women experience accelerated arterial endothelial dysfunction with older age, which puts them at heightened risk of cardiovascular diseases. Postmenopausal women demonstrated the elevated production of the RONS responsible for the intravascular oxidation of LDLs and the tyrosine nitration of proteins. However, RONS generation promoted the endothelial regeneration process expressed by considerable changes in the circulating EPCs and HMBG1. The values for the inflammatory-specific variables, such as the NLR, the SII, the NHR, and the CHR, were reduced in the women and showed their diagnostic utility for clinical prognosis in aging-related vascular dysfunction. Notably, other studies have shown that sex-dependent differences in cardiovascular health and disease should gain more attention in the scientific arena. It should be emphasized that although the evidence supports the role of sex differences in precision medicine, the European Society of Cardiology Guidelines from 2018–2023 do not sufficiently account for gender-specific medicine [89].

## Figures and Tables

**Figure 1 ijms-25-12203-f001:**
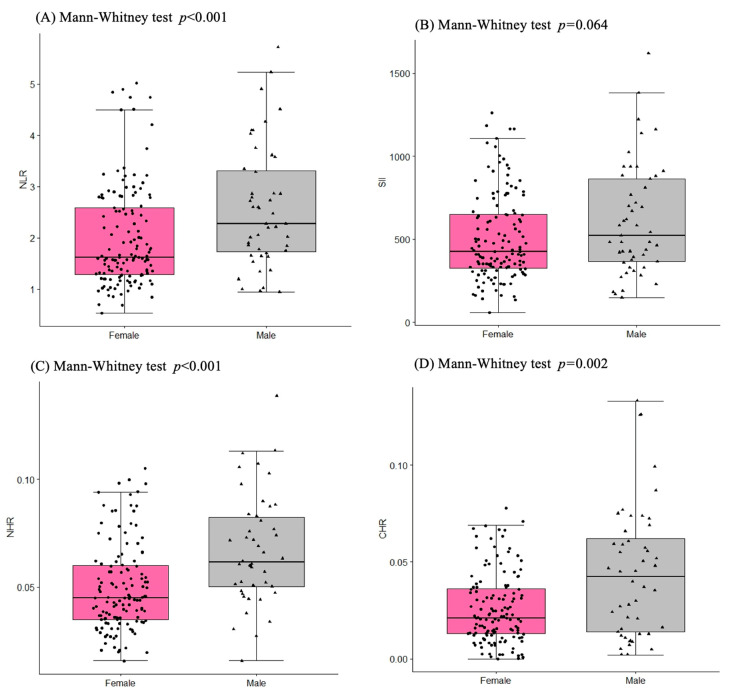
Inflammatory-specific variables. (**A**) NLR, neutrophil/lymphocyte ratio; (**B**) SII, systemic immune inflammation index; (**C**) NHR, neutrophils/HDL ratio; (**D**) CHR, C-reactive protein/HDL ratio.

**Figure 2 ijms-25-12203-f002:**
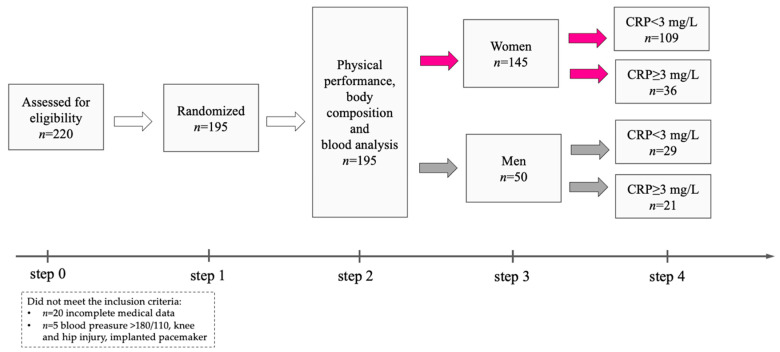
Study flow diagram; CRP, C-reactive protein.

**Table 1 ijms-25-12203-t001:** Basic characteristics of study sample.

	Female *n* = 145		Male *n* = 50		*p* Value
	Mean ± SD	Me (Min; Max)	Mean ± SD	Me (Min; Max)	
Age (years)	70.6 ± 6.7	69.0 (60.0; 93.0)	77.3 ± 8.0	77.5 (66.0; 96.0)	<0.001
Weight (kg)	67.9 ± 8.9	67.6 (44.8; 97.2)	79.2 ± 12.4	78.3 (44.8; 102.0)	<0.001
Height (cm)	159.4 ± 5.5	159.0 (143.0; 172.5)	168.0 ± 6.9	168.6 (150.0; 179.5)	<0.001
BMI (kg/m^2^)	26.8 ± 3.5	26.8 (18.6; 38.8)	28.1 ± 3.9	28.3 (19.9; 40.8)	0.031
VFC (VF unit)	9.7 ± 1.9	9.0 (5.0; 15.0)	15.6 ± 3.6	16.0 (8.0; 25.0)	<0.001
FM (kg)	24.2 ± 6.1	23.7 (8.8; 48.7)	20.4 ± 6.7	21.3 (7.7; 36.0)	0.002
FM%	34.9 ± 5.2	35.2 (17.4; 45.6)	25.0 ± 5.7	25.7 (12.1; 35.6)	<0.001
FFM (kg)	43.9 ± 4.5	43.5 (33.7; 63.3)	59.0 ± 7.2	59.9 (36.6; 73.0)	<0.001
SBP (mmHg)	146.5 ± 20.5	144.0 (95.0; 213.0)	147.5 ± 20.6	149.0 (99.0; 195.0)	0.324
DBP (mmHg)	80.9 ± 11.7	79.0 (55.0; 119.0)	82.3 ± 12.6	81.5 (55.0; 109.0)	0.210
6MWT (m)	454 ± 75	450 (265; 720)	438 ± 64	445 (320; 570)	0.363
Gait speed (m/s)	1.3 ± 0.2	1.3 (0.7; 2.0)	1.2 ± 0.2	1.2 (0.9; 1.6)	0.354

Abbreviations: BMI, body mass index; VFC, visceral fat content; FM, fat mass; FFM, fat-free mass; SBP, systolic blood pressure; DBP, diastolic blood pressure; 6MWT, 6 min walk test; SD, standard deviation; Me, median; min, minimum; max, maximum.

**Table 2 ijms-25-12203-t002:** Hematological variables.

	Female *n* = 145		Male *n* = 50		*p* Value
	Mean ± SD	Me (Min; Max)	Mean ± SD	Me (Min; Max)	
WBC (10^3^/µL)	6.25 ± 2.00	5.86 (2.90; 5.73)	6.65 ± 1.89	6.52 (2.48; 13.36)	0.116
Neutrophils (10^3^/µL)	4.00 ± 1.58	3.60 (1.52; 10.61)	4.45 ± 1.57	4.06 (1.75; 9.93)	0.012
Lymphocytes (10^3^/µL)	2.12 ± 0.79	2.04 (0.20; 5.63)	1.83 ± 0.66	1.77 (0.39; 3.42)	0.013
Monocytes (10^3^/µL)	0.45 ± 0.21	0.40 (0.07; 1.25)	0.43 ± 0.26	0.38 (0.10; 1.69)	0.356
Platelets (10^3^/µL)	260 ± 64	255 (37; 469)	226 ± 54	214 (125; 346)	<0.0001
RBC (10^3^/µL)	4.70 ± 0.42	4.76 (2.90; 5.73)	4.71 ± 0.77	4.66 (2.88; 8.28)	0.843
Hb (g/dL)	13.50 ± 3.87	13.65 (8.80; 16.10)	13.59 ± 1.56	13.45 (9.20; 18.60)	0.864
Hct%	38.28 ± 3.21	38.60 (26.82; 46.58)	38.31 ± 5.17	37.82 (23.63; 56.56)	0.996
MCV fL	81.67 ± 4.09	81.00 (68.0; 97.0)	81.68± 4.56	80.00 (68.0; 96.0)	0.252
MCH (pg/RBC)	28.79 ± 1.47	28.75 (24.40; 35.90)	29.09 ± 1.77	29.00 (22.50; 32.70)	0.164
MCHC (g/dL)	35.28 ± 1.00	35.40 (31.10; 38.10)	35.59 ± 1.27	35.65 (32.30; 38.90)	0.045
RDW%	13.06 ± 1.14	12.90 (27.70; 59.20)	13.54 ± 2.29	13.15 (11.60; 27.80)	0.012

Abbreviations: WBC, white blood cell; MPV, mean platelet volume; RBC, red blood cell; Hb, hemoglobin; Hct, hematocrit; MCV, mean cell volume; MCH, mean corpuscular hemoglobin; MCHC, mean corpuscular hemoglobin concentration; RDW, red cell distribution width; SD, standard deviation; Me, median; min, minimum; max, maximum.

**Table 3 ijms-25-12203-t003:** Lipoprotein–lipid profile.

	Female *n* = 145		Male *n* = 50		*p* Value
	Mean ± SD	Me (Min; Max)	Mean ± SD	Me (Min; Max)	
TG (mg/dL)	138.26 ± 46.32	132.31 (44.72; 381.40)	120.55 ± 46.17	116.16 (36.90; 271.98)	<0.001
TC (mg/dL)	240.16 ± 52.29	236.14 (129.40; 421.75)	193.47 ± 44.57	193.60 (105.00; 298.75)	<0.001
LDL (mg/dL)	86.51 ± 29.96	82.41 (33.22; 210.44)	91.39 ± 34.56	90.00 (26.08; 178.24)	0.375
HDL (mg/dL)	78.01 ± 16.59	77.94 (22.30; 154.02)	64.30 ± 17.88	60.35 (29.70; 108.25)	<0.001
non-HDL (mg/dL)	162.69 ± 53.88	154.99 (61.50; 344.80)	129.16 ± 41.37	121.35 (36.20; 235.04)	<0.001
oxLDL (mg/dL)	0.066 ± 0.052	0.066 (0.004; 0.240)	0.019 ± 0.014	0.016 (0.003; 0.075)	<0.001
TG/HDL	1.926 ± 1.157	1.645 (0.575; 9.394)	2.188 ± 1.138	2.052 (0.702; 5.474)	0.435

Abbreviations: TG, triglyceride; TC, total cholesterol; LDL, low-density lipoprotein; HDL, high-density lipoprotein; non-HDL, cholesterol calculated by subtracting the HDL value from the TC; oxLDL, oxidized low-density lipoprotein; SD, standard deviation; Me, median; min, minimum; max, maximum.

**Table 4 ijms-25-12203-t004:** Endothelium-specific variables.

	Female *n* = 145		Male *n* = 50		*p* Value
	Mean ± SD	Me (Min; Max)	Mean ± SD	Me (Min; Max)	
H_2_O_2_ (ng/mL)	672 ± 573	389 (171; 1988)	359 ± 103	297 (273; 588)	0.014
NO (μmol/L)	331 ± 250	226 (19; 1006)	135 ± 43	138 (36; 211)	<0.001
3-NitroT (nmol/mL)	1.88 ± 1.84	1.00 (0.60; 9.08)	1.01 ± 0.50	0.74 (0.62; 2.21)	<0.001
EPCs (ng/mL)	19.53 ± 16.43	11.67 (3.57; 70.55)	11.81 ± 7.39	10.73 (3.22; 50.12)	0.396
HMGB1 ng/mL	50.77 ± 19.15	43.98 (28.48; 98.74)	39.58 ± 9.84	38.63 (28.15; 58.89)	0.099

Abbreviations:; NO, nitric oxide; 3-NitroT, 3-nitrotyrosine; H_2_O_2_, hydrogen peroxide; HMGB1, high-mobility group box 1; EPCs, endothelial progenitor cells; SD, standard deviation; Me, median; min, minimum; max, maximum.

**Table 5 ijms-25-12203-t005:** The statistical characteristics of the ROC curve for the univariate logistic model.

	AUC	Cut-Off	Specificity (%)	Sensitivity (%)	OR	95% CI	*p* Value
NLR	0.655	1.67	77.2	53.3	3.87	1.911–7.829	<0.001
SII	0.627	350	86.0	34.1	3.17	1.383–7.244	0.003
NHR	0.636	0.064	52.6	77.6	3.82	1.991–7.451	0.004
CHR	0.980	0.038	94.7	93.3	252	65–967	<0.001

Abbreviations: AUC, the area under the curve; cut-off, the optimal threshold value for clinical stratification; OR, odds ratio; 95% CI, confidence interval.

## Data Availability

The raw data supporting the conclusions of this article will be made available by the authors without undue reservation.

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
