# Peer review of "Endothelial Dysfunction with Aging: Does Sex Matter?"

_ijms, 2024, doi:10.3390/ijms252212203_

Round 1
Reviewer 1 Report
Comments and Suggestions for Authors
The submitted paper investigates the role of oxidative stress, inflammation, and their interaction with endothelial dysfunction, particularly in older adults, with a focus on sex-specific differences. The study involved 195 participants (145 women, 50 men, average age 72) and examined how the production of reactive oxygen and nitrogen species (RONS) affects vascular health. Please find below specific comments:
- Since all women are menopausal I would be more specific with the title (e.g. endothelial dysfunction in older adults: does sex matter?). Furthermore, it appears that on average this is a cohort of hypertensive individuals. This is an important information to add.
- Authors should disclose why men are underrepresented compared to women.
- It is of crucial importance to disclose lipid lowering therapies, why this information is not provided?
- Since most risk calculators are based on calculated LDL instead of directly dosed, it would be useful to include calculated LDL differences (e.g. with Sampson equation and other novel equations in addition to Friedewald)
- I would omit reference values for hematological variables.
- Sex differences in cardiovascular health and disease should gain more attention in the scientific arena. It must be noted that however evidence supports the role of sex differences in precision medicine, ESC guidelines do not sufficiently account for gender medicine (see doi: 10.3390/jcm13144026.)
Author Response
Review 1
We greatly appreciate your time and effort dedicated to providing feedback on our manuscript and we are grateful for the insightful comments on and valuable improvements to our paper. All the suggestions helped us to evaluate our outcomes even more precisely in order to deliver improved, high quality scientific manuscript which we hope will now meet the high standards of International Journal of Molecular Sciences.
Comments and Suggestions for Authors
The submitted paper investigates the role of oxidative stress, inflammation, and their interaction with endothelial dysfunction, particularly in older adults, with a focus on sex-specific differences. The study involved 195 participants (145 women, 50 men, average age 72) and examined how the production of reactive oxygen and nitrogen species (RONS) affects vascular health. Please find below specific comments:
1/ Since all women are menopausal, I would be more specific with the title (e.g. endothelial dysfunction in older adults: does sex matter?). Furthermore, it appears that on average this is a cohort of hypertensive individuals. This is an important information to add.
Thank you for your comment. We followed the WHO classification to distinguish the age ranges that define old age. Our study population represented both groups of young elderly (aged 60 to 75 years) and old elderly (aged 75-89). Therefore, our intention was to indicate and emphasize the aging process without indicating a specific age group as our study was not confined to just one age range. The term “older adults” has been included into the Abstract section and Keywords.
As hypertension is an increasingly recognized chronic disease, and the number of cases increases with age, we included information that 84% of the study population was on medication for hypertension in subsection 4.1 Study population of the Materials and Methods.
2/ Authors should disclose why men are underrepresented compared to women.
In the Polish elderly population, we can observe large differences in life expectancy. According to the Statistics Poland GUS 2022 - the average life expectancy for men is 74.7 years and for women 82.0 years [https://stat.gov.pl/en/topics/population/life-expectancy/life-expectancy-of-poland-in-2023,2,17.html], which determines the differences in numbers in sex groups. Another problem is that older men are considerably more reluctant to participate in medical tests (screening), which is also related to their low care for their health.
The sample size calculation was performed before the subject’s recruitment. Considering the total number of people attending the University of the Third Age (n=220), a confidence level of 95% and a margin of error of 5%, the optimal sample size should be at least n=141 people. After applied the inclusion and exclusion criteria and considering the sample size, eventually, one hundred and ninety-five participants aged 72.2 ± 7.8 years (females n=145, males n=50) were included in the project.
3/ It is of crucial importance to disclose lipid lowering therapies, why this information is not provided?
Thank you for pointing this out. Patients had a history of whether or not they were taking lipid-lowering medications. In addition, patients do not regularly follow the recommendations of the medications they are taking and dietary mistakes contribute to lipid disorders. We supplemented subsection 4.1 Study population of Materials and Methods with the information that the interview and medical records revealed that 14% of patients participating in the study were taking lipid-lowering medications.
4/ Since most risk calculators are based on calculated LDL instead of directly dosed, it would be useful to include calculated LDL differences (e.g. with Sampson equation and other novel equations in addition to Friedewald)
We can assure you that serum LDL measurement was carried out directly and automatically using BM200 Biomaxima equipment and dedicated Biomaxima kit No 1-056-0060 (Lublin, Poland).
5/ I would omit reference values for hematological variables.
Thank you for your remark. Tables 2 and 3 have been corrected as suggested.
6/ Sex differences in cardiovascular health and disease should gain more attention in the scientific arena. It must be noted that however evidence supports the role of sex differences in precision medicine, ESC guidelines do not sufficiently account for gender medicine (see doi: 10.3390/jcm13144026.)
Thank you very much for this valuable suggestion, we have used the indicated article to enrich the last Conclusion section.

Reviewer 2 Report
Comments and Suggestions for Authors
Jakub Jozue Wojtacha et al. attempted to demonstrate potential association based sex-specific changes in oxi-inflammatory response with endothelial function in elderly subjects. The topic is interesting and might have some clinical relevance with identification of novel biomarkers.
1/ The size of population is relatively small and not that well balanced in size among groups (female 145 vs. male 50) with some differences in average age in each group (female 70y vs. male 77y). This should be taken in consideration to draw statistical significance analysis.
2/ Fig. 2 (Study chart): Did the authors try to analyze relevant data in stratified groups based on CRP levels (CRP<3 mg/dL vs. CRP>=3 mg/dL) in both male and female subjects?
3/ Table 3 & 4: Values were given as Mean +/- SD as well as medians. Range for lowest and highest values should be given for median values.
4/ Please provide reference for the following statement: "The excess of superoxide anion reacts with NO to produce peroxynitrite (ONOO) whose level could increase even 1000000-fold".
5/ Did the authors attempt to measure levels of nitrated lipoproteins such as HDL and LDL? if yes how was the correlation looks like between nitrated LDL/HDL with other biomarkers among subjects?
Author Response
Review 2
We greatly appreciate your time and effort dedicated to providing feedback on our manuscript and we are grateful for the insightful comments on and valuable improvements to our paper. All the suggestions helped us to evaluate our outcomes even more precisely in order to deliver improved, high quality scientific manuscript which we hope will now meet the high standards of International Journal of Molecular Sciences.
Comments and Suggestions for Authors
Jakub Jozue Wojtacha et al. attempted to demonstrate potential association-based sex-specific changes in oxi-inflammatory response with endothelial function in elderly subjects. The topic is interesting and might have some clinical relevance with identification of novel biomarkers.
1/ The size of population is relatively small and not that well balanced in size among groups (female 145 vs. male 50) with some differences in average age in each group (female 70y vs. male 77y). This should be taken in consideration to draw statistical significance analysis.
Thank you for your comment. In the Polish elderly population, we can observe large differences in life expectancy. According to the Statistics Poland GUS 2022 - the average life expectancy for men is 74.7 years and for women 82.0 years [https://stat.gov.pl/en/topics/population/life-expectancy/life-expectancy-of-poland-in-2023,2,17.html], which determines the differences in numbers in gender groups. Another problem is that older men are considerably more reluctant to participate in medical tests (screening), which is also related to their low care for their health.
The sample size calculation has been performed before the subjects recruitment. Considering the total number of people attending the University of the Third Age (n=220), a confidence level of 95% and a margin of error of 5%, the optimal sample size should be at least n=141 people. After applied the inclusion and exclusion criteria and considering the sample size, eventually, one hundred and ninety-five participants aged 72.2 ± 7.8 years (females n=145, males n=50) were included in the project.
2/ Fig. 2 (Study chart): Did the authors try to analyze relevant data in stratified groups based on CRP levels (CRP<3 mg/dL vs. CRP>=3 mg/dL) in both male and female subjects?
Yes, we stratified groups based on CRP levels (CRP<3 mg/dL vs. CRP ≥3 mg/dL) in both male and female subjects. In two tables below we present the most important variables regarding important factors of endothelial dysfunction for CRP <3 mg/L and CRP ≥3 mg/L group.
TG, TC, HDL and non-HDL ​​differed significantly between the sexes in low- and high-grade inflammatory status but NLR and CHR did not. We rejected the possibility of presenting these results mainly due to the small number of women and men in stratified CRP groups. The explanation has been provided in section 2.3. Lipoprotein-lipid profile :“There were significant differences recorded for TG, TC, HDL and non-HDL but not for other lipoprotein-lipid and inflammatory variables in women and men stratified based on CRP levels.”
Important factors of endothelial dysfunction in group CRP <3 mg/L
Important factors of endothelial dysfunction in group CRP ≥3 mg/L
|
|
Female n=109 |
Male n=29 |
p value |
|||
|
Mean ± SD |
Me (min, max) |
Mean ± SD |
Me (min, max) |
|||
|
NLR |
2.41 ± 2.17 |
1.60 (0.53; 14.93) |
2.54 ± 1.27 |
2.23 (0.97; 5.90) |
0.010 |
|
|
CHR |
0.019 ± 0.011 |
0.017 (0.000; 0.058) |
0.022 ± 0.016 |
0.016 (0.002; 0.059) |
0.475 |
|
|
TG mg/dL |
133 ± 43 |
130 (45; 380) |
111 ± 32 |
115 (38; 169) |
0.003 |
|
|
TC mg/dL |
238 ± 51 |
235 (129; 422) |
193 ± 42 |
197 (105; 299) |
0.001 |
|
|
LDL mg/dL |
86.05 ± 29.46 |
82.16 (45.42; 210.44) |
86.72 ± 30.93 |
75.88 (26.08; 160.54) |
0.971 |
|
|
HDL mg/dL |
77.07 ± 16.65 |
76.95 (0.00; 121.24) |
67.51 ± 18.64 |
64.33 (29.70; 108.25) |
0.001 |
|
|
Non-HDL mg/dL |
161 ± 53 |
153 (62; 345) |
125 ± 39 |
121 (36; 235) |
0.001 |
|
|
|
Female n=36 |
Male n=21 |
p value |
||
|
Mean ± SD |
Me (min, max) |
Mean ± SD |
Me (min, max) |
||
|
NLR |
2.93 ± 1.93 |
2.24 (1.13; 8.92) |
3.70 ± 3.04 |
2.73 (1.53; 15.33) |
0.174 |
|
CHR |
0.067 ± 0.032 |
0.057 (0.023; 0.180) |
0.093 ± 0.056 |
0.074 (0.050; 0.282) |
0.003 |
|
TG mg/dL |
155 ± 50 |
142 (93; 381) |
134 ± 58 |
117 (37; 272) |
0.007 |
|
TC mg/dL |
247 ± 56 |
250 (144; 394) |
194 ± 48 |
192 (110; 267) |
0.001 |
|
LDL mg/dL |
86.63 ± 33.65 |
84.84 (5.87; 187.88) |
97.83 ± 38.10 |
93.64 (48.22; 178.24) |
0.265 |
|
HDL mg/dL |
78.71 ± 20.67 |
78.56 (40.60; 154.02) |
59.87 ± 15.72 |
58.40 (40.30; 91.55) |
0.001 |
|
Non-HDL mg/dL |
167.97 ± 56.09 |
170.18 (84.70; 342.04) |
134.53 ± 43.38 |
121.42 (58.70; 206.60) |
0.025 |
3/ Table 3 & 4: Values were given as Mean +/- SD as well as medians. Range for lowest and highest values should be given for median values.
Following the Reviewer’s suggestion, Tables 1-4 have been corrected.
4/ Please provide reference for the following statement: "The excess of superoxide anion reacts with NO to produce peroxynitrite (ONOO) whose level could increase even 1000000-fold".
Please accept the apologies for the editorial error. The paragraph has been corrected and supplemented with two references.
“The excess of superoxide anion reacts with NO to produce peroxynitrite (ONOO) which leads to comparable redox changes of other oxidants but is regarded 1000-fold more potent in causing these redox reactions [https://pubmed.ncbi.nlm.nih.gov/36719770/]. ONOO, in turn, can “uncouple” endothelial NO synthase to become a dysfunctional superoxide-generating enzyme that contributes to vascular oxidative stress [https://pubmed.ncbi.nlm.nih.gov/20306272/].”
5/ Did the authors attempt to measure levels of nitrated lipoproteins such as HDL and LDL? if yes how was the correlation looks like between nitrated LDL/HDL with other biomarkers among subjects?
Thank you for Your suggestion. In fact, we did not measure the levels of nitrated lipoproteins HDL and LDL in this project, but we are planning to expand the scope of the research by commercially available ELISA kits for NT-HDL and NT-LDL recommend by Adedayo et al. [https://pubmed.ncbi.nlm.nih.gov/33049686/].

Round 2
Reviewer 2 Report
Comments and Suggestions for Authors
The authors have satisfactorily addressed all my concerns.